# Exploiting induced pluripotent stem cell-derived macrophages to unravel host factors influencing *Chlamydia trachomatis* pathogenesis

Amy T.Y. Yeung[1], Christine Hale[1], Amy H. Lee[2], Erin E. Gill[2], Wendy Bushell[1], David Parry-Smith[1], David Goulding[1], Derek Pickard[1], Theodoros Roumeliotis[1], Jyoti Choudhary[1], Nick Thomson[1], William C. Skarnes[1], Gordon Dougan[1,3] & Robert E.W. Hancock[1,2]

*Chlamydia trachomatis* remains a leading cause of bacterial sexually transmitted infections and preventable blindness worldwide. There are, however, limited *in vitro* models to study the role of host genetics in the response of macrophages to this obligate human pathogen. Here, we describe an approach using macrophages derived from human induced pluripotent stem cells (iPSdMs) to study macrophage–*Chlamydia* interactions *in vitro*. We show that iPSdMs support the full infectious life cycle of *C. trachomatis* in a manner that mimics the infection of human blood-derived macrophages. Transcriptomic and proteomic profiling of the macrophage response to chlamydial infection highlighted the role of the type I interferon and interleukin 10-mediated responses. Using CRISPR/Cas9 technology, we generated biallelic knockout mutations in host genes encoding IRF5 and IL-10RA in iPSCs, and confirmed their roles in limiting chlamydial infection in macrophages. This model can potentially be extended to other pathogens and tissue systems to advance our understanding of host-pathogen interactions and the role of human genetics in influencing the outcome of infections.

[1] Wellcome Trust Sanger Institute, Wellcome Trust Genome Campus, Hinxton, Cambridge CB10 1SA, UK. [2] Centre for Microbial Diseases and Immunity Research, University of British Columbia, Vancouver, British Columbia, Canada V6T 1Z4. [3] Department of Medicine, Addenbrookes Hospital, Box 157, Hills Rd, Cambridge CB2 0QQ, UK. Correspondence and requests for materials should be addressed to R.E.W.H. (email: bob@hancocklab.com).

Chlamydia trachomatis is a leading cause of bacterial sexually transmitted diseases (STDs) worldwide, with more than a hundred million new cases of STDs caused by this pathogen annually. These Gram-negative obligate intracellular bacteria cause genital infections leading to pelvic inflammatory disease and infertility[1]. Chlamydia are also a leading cause of preventable blindness due to trachoma, as well as being the aetiological agents responsible for non-gonococcal urethritis, and Chlamydia-induced reactive arthritis. During arthritis, C. trachomatis likely reaches the joint from the urogenital system through circulating monocytes and macrophages. Indeed, although C. trachomatis preferentially infects epithelial cells, within the synovial tissues of reactive arthritis patients, viable, metabolically active Chlamydia predominately reside and persist within macrophages[2–5]. Although macrophages are not always a permissive environment for Chlamydia, their capacity to infect macrophages varies between strains, the source and subtype of macrophages and the multiplicity of infection (MOI) used for infection[6]. For example, Chlamydia does not grow in inflammatory M1 macrophages but grows well in M2 and M0 (resting) macrophages[7,8].

Consequently, macrophages play critical roles in defence against chlamydial infections. For example, upon recruitment to sites of infection, the ability of macrophages to phagocytose and kill Chlamydia is crucial to limiting the progression of the infection. Moreover, the infiltration of macrophages into infected tissues can lead to inflammation, a phenotype commonly associated with Chlamydia infections[1]. Conversely, chlamydial persistence in macrophages can also contribute to chronic inflammation and delays in the efficacy of antibiotic treatment. Therefore, an understanding of the mechanisms that control chlamydial survival in human macrophages could have major implications for the development of therapeutic strategies.

Despite the importance of human genetics in the complex interactions between Chlamydia and human macrophages, the availability of cell-based human systems to study this interplay is limited. Most in vitro models currently used to study Chlamydia–macrophage interactions involve macrophages derived from other mammals, immortalized macrophage-lineage cell lines or primary monocytes/macrophages, predominantly from mice[9,10]. Aside from clear differences in the bacterial response of macrophages from humans and mice, immortalized cell lines are also far removed from the normal genetic state, including harbouring extensive genome rearrangements[11]. Moreover, primary monocytes and macrophages are inherently difficult to genetically manipulate, thus limiting the ability to study how macrophage genetics impact on Chlamydia pathogenesis.

Induced pluripotent stem-cell (iPSC) derived macrophages (iPSdMs) can potentially overcome some of the drawbacks of classical cell lines and can be used as an alternative source of primary human macrophages, providing a more personalised and tractable genetic approach. Human iPSCs can be readily genetically manipulated, using CRISPR/Cas9 technology, or isolated from individuals with common or rare disease-associated genotypes.

Here we differentiated human iPSCs into macrophages and showed, through a combination of flow cytometric analyses, microscopy, transcriptomic and proteomic profiling that human iPSdMs respond to C. trachomatis in a similar manner to macrophages derived from human blood monocytes. Using CRISPR/Cas9 genome editing approach[12], we generated human iPSC mutations in candidate genes identified from the functional genomic studies, to validate and identify novel mechanisms involved in limiting Chlamydia growth in macrophages.

## Results

**Formation of chlamydial inclusions inside infected iPSdMs.** We investigated whether human iPSdMs could support the productive growth of Chlamydia infection. To differentiate human iPSCs into macrophages, we adapted and modified the approach from Hale et al.[13] (Methods, Fig. 1a). Human iPSdMs derived from both the KOLF2 and FPS10C iPSC lines were generally morphologically indistinguishable from primary human blood monocyte-derived macrophages and expressed high levels of macrophage markers, including CD11b, CD14, EMR1 and CD68, while the stem cell pluripotency markers, including SSEA-4 and OCT-3/4, were correspondingly low (Fig. 1a–c). Transcriptomic analysis of naïve human KOLF2 iPSdMs compared to primary human blood monocyte-derived macrophages revealed > 95% of genes were similarly expressed in both cell types, in accordance with previous studies[14].

To facilitate visualization of the infection of human iPSdMs with Chlamydia, we used a C. trachomatis serovar F SWFP⁻ derivative, harbouring plasmid pGFP::SW2 that directs the constitutive expression of green fluorescent protein (GFP)[15]. At 1 h post infection of human iPSdMs and blood monocyte-derived macrophages with C. trachomatis/pGFP::SW2), the GFP fluorescence generated by the bacteria was too weak to detect by the Cellomics CellInsight NTX but transmission electron microscopy (TEM) revealed small inclusions containing single or a few Chlamydia elementary bodies (EBs) inside the infected macrophages (Fig. 2a). After 24 h of Chlamydia infection, Cellomics identified generally 5–7% of both human iPSdMs and blood monocyte-derived macrophages were infected. These infected macrophages revealed, via TEM, single or multiple large inclusions (Fig. 2b). Formation of the inclusions distorted the morphology of the macrophages, which had increased substantially in size compared to equivalent uninfected macrophages. By 48 h post infection, the infected macrophages had rounded up with some macrophages lysed, releasing EBs into the extracellular medium (Fig. 2c). Using Cellomics technology, we observed the average GFP intensity from C. trachomatis infected human iPSdMs and human blood monocyte-derived macrophages increased to a maximum at 48 h. After this time point, the GFP intensity gradually decreased (Fig. 2d). C. trachomatis progeny released at 48 h were infective, as the released EBs were able to re-infect uninfected macrophages and McCoy epithelial cells (Fig. 3a–c).

**Human iPSdM responses to C. trachomatis infection.** We next determined how human iPSdMs responded to C. trachomatis challenge using a combination of transcriptomics and proteomics. Prior to infection, investigation of the expression of key macrophage differentiation markers indicated that (unstimulated) iPSDMs expressed low to very low levels (0.1–5 counts per million (c.p.m.)) of M1 markers (CXCL10, CXCL11 and TAP1) and also modest levels (10–1,000 c.p.m.) of M2 markers (CCL13, CD209, F13A1, MRC1 and TGM2). This was consistent with previous studies on iPSDM with another cell lineage which concluded that iPSDMs are fully mature but can be differentiated into M2 or M1 polarized cells[13]. Thus these cells were considered M0 macrophages that, like M2 macrophages but not M1 macrophages, are permissive for Chlamydia growth[7,8].

At 24 h post infection, the genes dysregulated by Chlamydia in infected cf. uninfected iPSdM cells were compared to those observed in blood-derived macrophages. A total of 2,029 genes repeatedly changed expression after Chlamydia infection in both iPSdMs and blood-derived macrophages, with 1,194 genes upregulated and 835 gene downregulated (fold change of ≥ 1.5, $P \leq 0.05$; see Supplementary Data 1 for list of selected DE genes

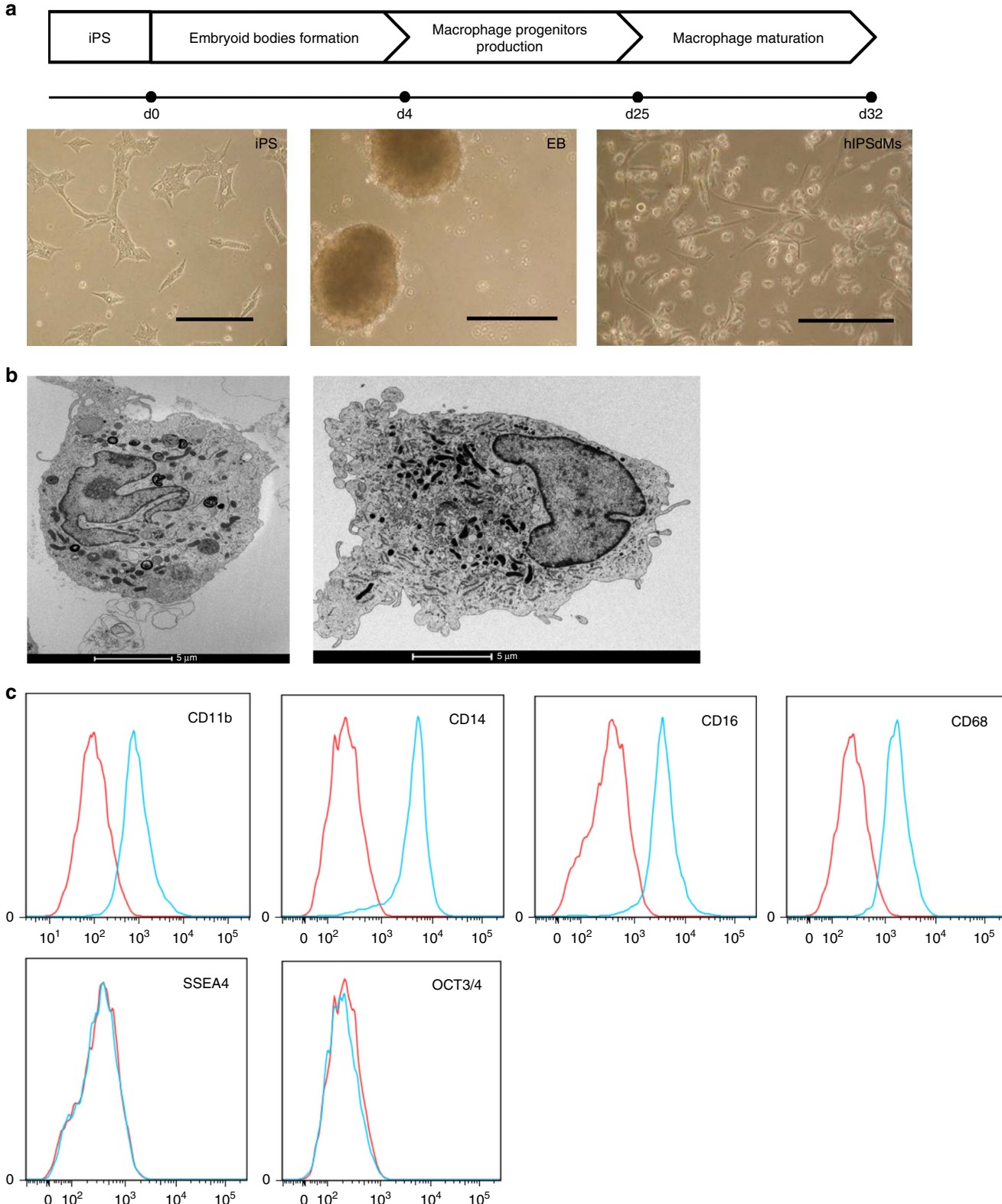

**Figure 1 | Differentiation of macrophages from human iPSCs.** (**a**) Schematic diagram of the method of *in vitro* differentiation from human iPSCs to macrophages, with accompanying representative light microscopy images for each stage. Scale bars, 200 μm. (**b**) EM images of a human iPSdM (left) and a human blood monocyte differentiated macrophage (right). (**c**) Flow cytometry analyses for the expression of macrophage markers (CD11b, CD14, CD16, CD68) and pluripotency markers (SSEA4, OCT3/4) on human iPSdMs. Red lines represent cells stained with control isotype and blue lines represent cells stained with the relevant antibody.

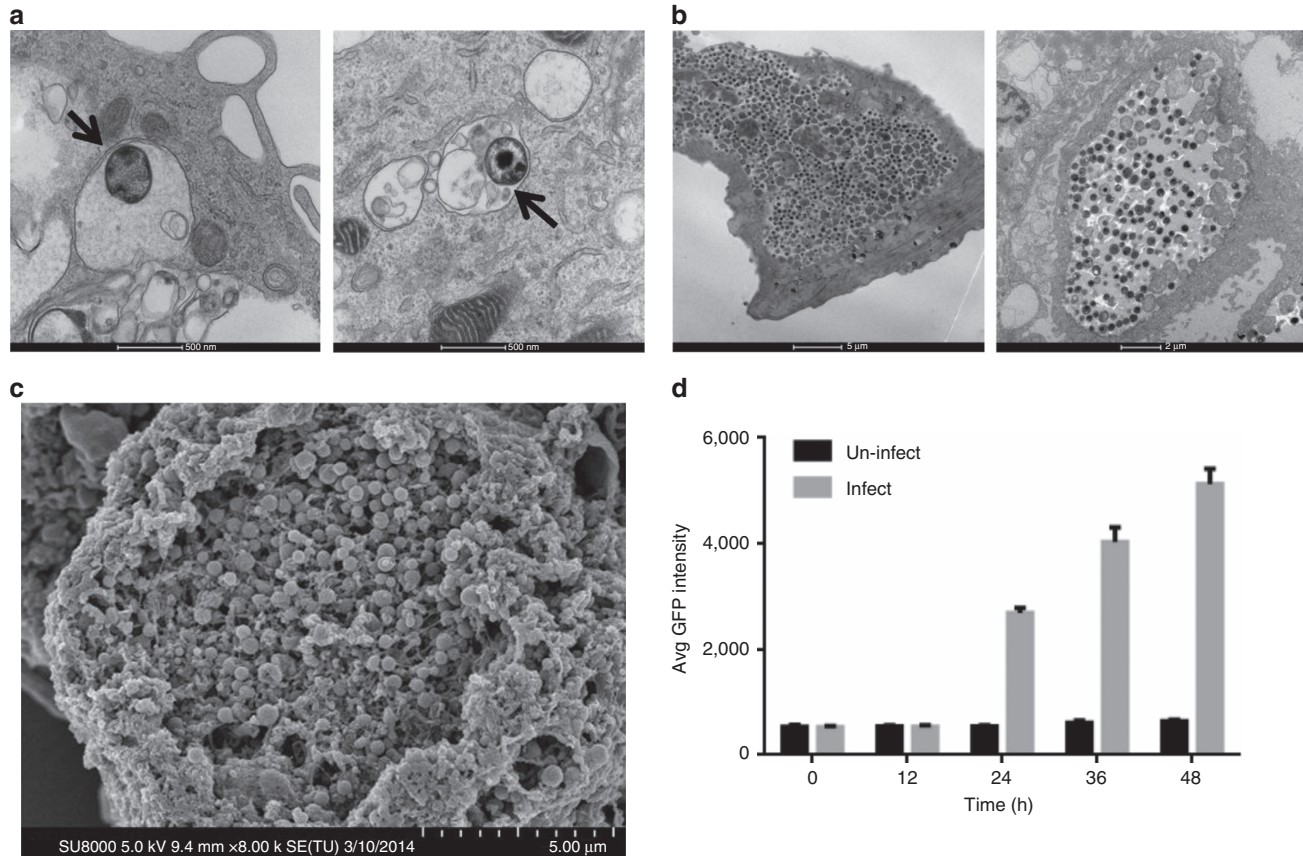

**Figure 2 | Imaging of human iPSdMs infected with *C. trachomatis*.** (**a**) Representative transmission electron microscopy (TEM) image of iPSdM (left) and blood monocyte-derived macrophage (right) infected for 1 h (arrows indicating *Chlamydia*-containing inclusions formed inside infected macrophages), (**b**) representative TEM image of iPSdM (left) and blood monocyte-derived macrophage (right) infected for 24 h. (**c**) Representative scanning electron microscopy image of iPSdM infected for 48 h. (**d**) Changes in GFP intensity during a 48 h infection of iPSdMs with GFP-tagged *C. trachomatis*. Results are the average of three independent measurements ± s.d. using the Incucyte imaging system.

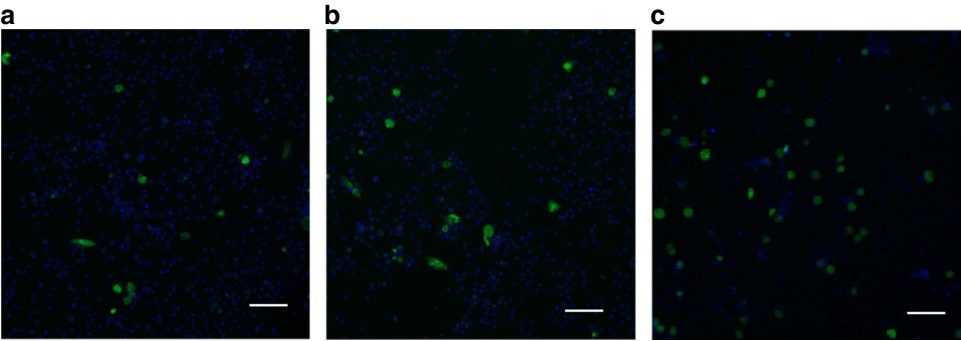

**Figure 3 | *In vitro* infection of mammalian cells with GFP-tagged *C. trachomatis* harvested from 48 h infections.** (**a**) Human iPSdMs. (**b**) Human blood monocyte-derived macrophages. (**c**) McCoy epithelial cells. Representative images were taken using the Cellomics CellInsight NXT at ×10 magnification, scale bar, 80 μm.

with the full data set provided as described in the 'Data availability' section in Methods). The promoters of these dysregulated genes frequently included, according to oPossum (http://opossum.cisreg.ca/oPOSSUM3/; z score >4), binding sites for the immune system transcription factors NFKB subunits p65, p50 and C-Rel, AP1, IRF1, IRF2, STAT1 and STAT3.

Pathway analysis of dysregulated genes in iPdSMs and blood macrophages following *Chlamydia* infection was undertaken using Sigora[16] that limits the repetitive assignment of the same

genes to multiple overlapping pathways. As shown in Table 1 (more complete version in Supplementary Data 2–4, including separate Reactome pathway analysis of up and downregulated genes), there was a strong over representation in infected cells of major immune-related pathways reflecting a strong interferon α, β and γ response. Dysregulation of pathway gene expression was also evident for various Toll-like receptor (TLR), IL-6, RIG-I and TRIF (MyD88-independent) pathways, the endosomal/vaculolar pathway, inhibition of translation, energy metabolism

**Table 1 | Selected pathways dysregulated upon chlamydial infection in iPSdMs and human blood monocyte-derived macrophages as shown by RNA-seq.**

| Pathway description | Corrected $P$ value |
| --- | --- |
| Interferon α/β signalling | 4.3E − 131 |
| Endosomal/vacuolar pathway | 3.01E − 70 |
| Eukaryotic translation elongation | 2.14E − 63 |
| Nonsense mediated decay independent of the exon junction complex | 3.35E − 28 |
| Platelet degranulation | 1.74E − 25 |
| Interferon γ signalling | 2.46E − 21 |
| TRIF-mediated TLR3/TLR4 signalling | 1.67E − 20 |
| Interleukin-6 signalling | 2.24E − 19 |
| Signalling by interleukins | 7.57E − 17 |
| Infectious disease | 3.10E − 16 |
| Metabolism of amino acids and derivatives | 1.05E − 15 |
| Negative regulators of RIG-I/MDA5 signalling | 2.41E − 13 |
| Glycosaminoglycan metabolism | 2.50E − 12 |
| Diseases of immune system | 2.28E − 11 |
| SRP-dependent cotranslational protein targeting to membrane | 7.30E − 10 |
| Interferon signalling | 1.37E − 09 |
| Peptide chain elongation | 1.98E − 08 |
| Respiratory electron transport, ATP synthesis by chemiosmotic coupling, and heat production by uncoupling proteins. | 1.48E − 06 |
| Activation of the mRNA upon binding of the cap-binding complex and eIFs, and subsequent binding to 43S | 3.89E − 06 |
| Downstream signalling of activated FGFR4 | 1.54E − 05 |
| Influenza infection | 1.78E − 05 |
| Gap junction trafficking and regulation | 4.51E − 05 |
| Signalling by NGF | 4.63E − 05 |
| Antigen processing-cross presentation | 7.44E − 05 |
| Metabolism of nucleotides | 1.51E − 04 |
| GTP hydrolysis and joining of the 60S ribosomal subunit | 1.69E − 04 |
| Regulation of IFNG signalling | 2.83E − 04 |
| Cell junction organization | 3.95E − 04 |
| Influenza life cycle | 5.54E − 04 |
| MyD88:Mal cascade initiated on plasma membrane | 5.64E − 04 |
| BMAL1:CLOCK,NPAS2 activates circadian gene expression | 9.20E − 04 |
| Chaperonin-mediated protein folding | 4.66E − 03 |
| Growth hormone receptor signalling | 5.93E − 03 |
| Sema4D induced cell migration and growth-cone collapse | 8.14E − 03 |
| Signalling by FGFR2 | 1.67E − 02 |
| GLI3 is processed to GLI3R by the proteasome | 2.08E − 02 |
| Cellular senescence | 2.90E − 02 |
| Toll-like receptor TLR6:TLR2 cascade | 3.19E − 02 |

These data show a subset of common overrepresented pathways (determined using Sigora[18]) in both iPSdMs and human blood monocyte-derived macrophages infected with *C. trachomatis* for 24-h (in comparison with their uninfected counterparts), as identified by RNA-Seq. Pathways were selected based on their novelty and lack of overlap.

and metabolism of amino acids (especially tryptophan; Supplementary Data 2–4) and nucleotides. Consistent in part with this, at 24 h post *Chlamydia* infection, human iPSdMs were shown, using Luminex bead-based multiplex assays, to express high levels of the pro-inflammatory cytokines IL-6 and TNFα, as well as IL-1β, chemokine IL-8 and the anti-inflammatory cytokine IL-10 (Fig. 4).

RNA-Seq revealed that genes associated with Type I interferon signalling were the most significantly upregulated including key transcription factors such as the interferon regulatory factors IRF-1 and 7 (upregulated by 2- and 5-fold), signal transducer and activator of transcription genes (STAT)-1 and -2 (4- and 2-fold), and interferon stimulated genes such as ISG15 (6-fold). In addition IRF5 was significantly upregulated and qRT-PCR confirmed that the extent of upregulation was 3.4 ± 0.2. Consistent with these observations being functionallly important, the use of an anti-IFNAR1 antibody to block IFN α/β receptor resulted in an 85.5 ± 6.8% increase in *C. trachomatis* infection as compared to untreated cells (Fig. 5a).

Interestingly, we observed very substantial upregulation of IDO1 by >800-fold in infected human iPSdMs. IDO1 has been recently shown to play an important role in defense against *Chlamydia pneumoniae*[17] and consistent with this, treatment of human iPSdMs with INCB024360 (ref. 18) (a competitive inhibitor of IDO1) resulted in an 83.6 ± 20.9% increase in *C. trachomatis* infection as compared to untreated cells (Fig. 5b).

A variety of cell surface receptors were also upregulated including ICAM-1 (7-fold) and CD44 (7-fold)[19,20], integrin family members including α2, α6, αM, β2, β3, β7 and β8 (2- to 14-fold)[21] and syndecan SDC2 (4-fold)[22], also indicating that *Chlamydia* changes the intrinsic interactions of macrophages with other host elements.

Of the 835 genes downregulated in human iPSdMs at 24 h post *Chlamydia* infection, the top hits (Supplementary Data 4) were associated with gap junction trafficking and regulation (as observed previously[23]), as well as nonsense mediated decay and various translation processes, which might be involved in either the host cells attempting to control *Chlamydia* intracellular growth (as occurs with other pathogens[24]), or *Chlamydia* diverting cellular resources to its own growth. Pathway analysis also revealed the downregulation of other cellular processes during macrophage infection related to tubulin, and gap junction trafficking pathways.

We also observed a substantial number of uniquely expressed genes in both iPSDM and MDM, consistent with previous observations by others[14] and likely related in part to the different

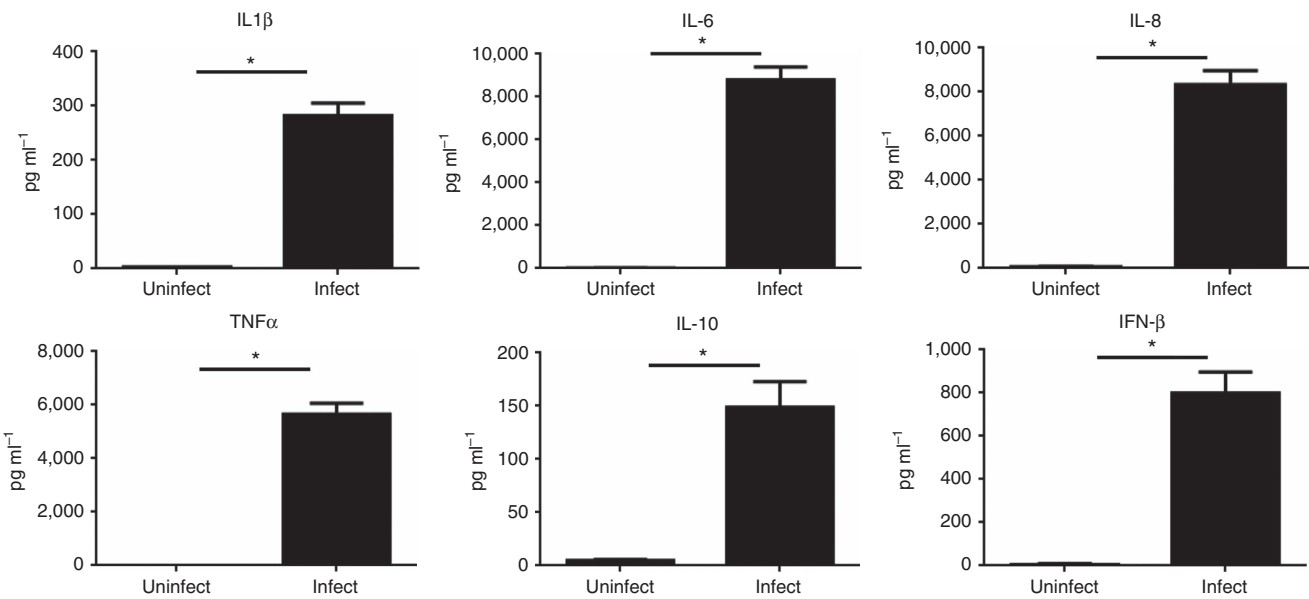

**Figure 4 | Overproduction of cytokines in human iPSdMs after infection with *C. trachomatis* for 24 h.** Results are the average of three independent measurements ± s.d. using the Luminex customised anti-human cytokine Milliplex kit. *Represents statistically significant different ($P < 0.05$) between uninfected and *Chlamydia*-infected as determined using two-way ANOVA.

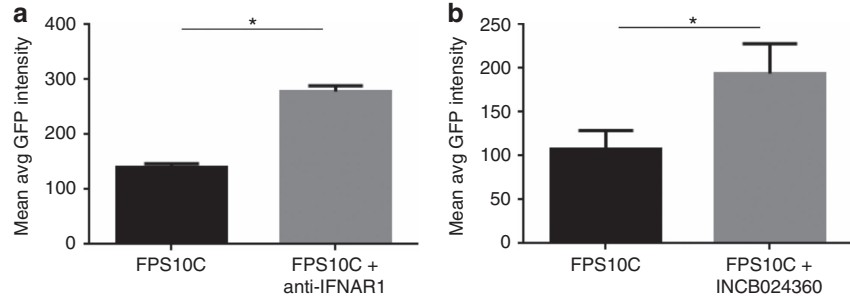

**Figure 5 | Effect of inhibitors on *C. trachomatis* infection of human iPSdMs. (a)** Anti-IFNAR1 antibody; **(b)** IDO inhibitor INCB024360. Results are the average of three independent measurements ± s.d. using the Cellomics CellInsight NXT. *Represent statistically significant different ($P < 0.05$) between untreated and treated as determined using two-way ANOVA.

derivation and culture methods (Supplementary Data 5–8). Many of these genes had modest read counts and/or did not fit into pathways, but Sigora enrichment analysis identified a number of iPSDM-unique pathways including extracellular matrix organization, collagen degradation, cell surface interactions at the vascular wall (Supplementary Data 5 and 6). Amongst MDM specific pathways (Supplementary Data 7 and 8), we observed certain cellular metabolic processes, including mitochondrial protein import, pyruvate metabolism, gluconeogenesis, as well as antigen processing and cell cycle regulation (G2/M transition and DNA repair).

To study the post-transcriptional host response during *C. trachomatis* infection of human iPSdMs at 24 h, we performed a detailed whole cell proteome study. At 24 h post infection, comparing infected to uninfected cells, proteomics revealed significant changes in 307 proteins with 229 upregulated and 78 downregulated (fold change (FC) of $>2.0$ or $<-2.0$ and $P$ value $<0.05$) (Supplementary Data 9 for a full list of DE proteins). The lower number of proteins found to be dysregulated may relate to the lower sensitivity of the proteomic methodologies. Up and downregulated proteins were separately submitted to the innate immunity interactome database and analysis

platform InnateDB, and over-representation (OR) analyses carried out. Of the 229 upregulated proteins, 79 pathways were found to be significantly over represented (Bonferroni corrected $P$ value $<0.05$) (Table 2 for selected OR upregulated pathways). The top upregulated pathways showed similarity to those identified by transcriptomics (indicated by an asterisk in Table 2), including interferon signalling and the inflammatory response, but several pathways were unique implying possible post-transcriptional regulation. Upregulated pathways identified through proteomics but not transcriptomics included lipoprotein/cholesterol metabolism (APOA1, APOE), and the complement and coagulation cascades (C4BPA, C7).

**Roles of IRF5 and IL10RA in *Chlamydia*–macrophage interaction.** To identify key host genes involved in macrophage–*Chlamydia* interactions, we used CRISPR/Cas technology to generate biallelic knockouts in human iPSCs (Methods, Fig. 6). To validate this methodology and uncover new insights, we explored the role of two candidate genes, not previously studied in human *Chlamydia* pathogenesis, based on our RNA-Seq/proteomics analyses, namely Interferon regulatory factor 5 (IRF5) and

**Table 2 | Pathways upregulated in human iPSdMs upon *C. trachomatis* infection as shown by proteomics.**

| Pathway name | Corrected *P* value |
| --- | --- |
| *Cytokine signalling in immune system | 6.4E − 14 |
| *Interferon signalling | 1.9E − 13 |
| *Interferon α/β signalling | 2.7E − 10 |
| *Interferon γ signalling | 1.5E − 07 |
| *RIG-I/MDA5 induction of IFN-α/β pathways | 7.5E − 04 |
| Cytokine-cytokine receptor interaction | 3.4E − 06 |
| *Chemokine receptors bind chemokines | 4.2E − 06 |
| *Immune system | 4.7E − 06 |
| *TRAF6 mediated NF-κB activation | 8.0E − 04 |
| *Interleukin-1 processing | 2.3E − 03 |
| *Tryptophan catabolism | 7.6E − 03 |
| *Chemokine signalling pathway | 1.2E − 03 |
| ISG15 antiviral mechanism | 1.7E − 03 |
| Complement and coagulation cascades | 1.7E − 03 |
| HDL-mediated lipid transport | 9.5E − 04 |
| IL12-mediated signalling events | 3.7E − 03 |
| Tryptophan metabolism | 5.4E − 03 |
| AP-1 transcription factor network | 8.0E − 03 |
| Lipoprotein metabolism | 7.7E − 03 |

These data show selected upregulated pathways in human iPSdMs infected with *C. trachomatis* for 24-h, in comparison with uninfected iPSdMs, as identified by proteomics. Pathways were selected based on their novelty and lack of overlap. The symbol * indicates pathways shown to be upregulated upon chlamydial infection by both proteomic and transcriptomic analyses.

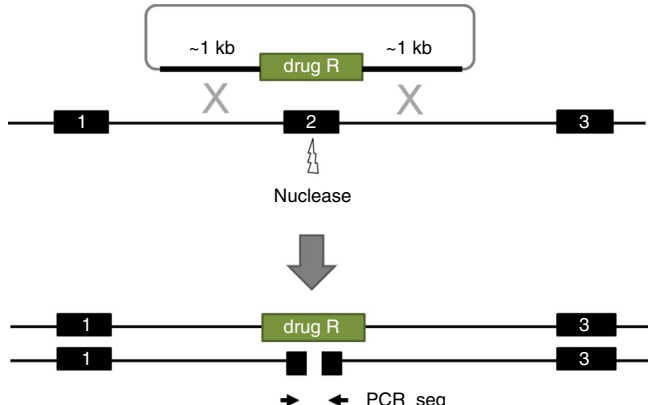

**Figure 6 | Schematic diagram for the generation of biallelic knockout mutants.** Strategy for the generation of biallelic knockouts is to replace a critical exon of one allele of the target gene with a drug selection cassette by homologous recombination and to screen clones for damage to the second allele induced by error-prone non-homologous end-joining (NHEJ). Since only one copy of the target exon will be present in correctly targeted clones, NHEJ-induced damage to the non-targeted allele can be simply assessed by Sanger sequencing of PCR products from the target exon.

Interleukin 10 receptor α (IL10RA). Human iPSCs lines harbouring homozygous mutations in each gene (two independently derived clones per gene to minimize the potential for off-target effects) were generated and subsequently differentiated into macrophages to investigate whether the candidate genes played roles in *C. trachomatis* infection. The bi-allelic mutation was validated for each mutant by PCR and sequencing (Supplementary Fig. 1A,B). Both mutant human iPS lines were able to differentiate into similar numbers of macrophages as the parental iPS line and expressed macrophage markers (CD11b, CD14, CD16, CD68 as measured by flow cytometry; Supplementary Fig. 1C). Moreover, all of the mutant iPSdMs showed an ability to phagocytose (Supplementary Fig. 1D).

Due to the apparent importance of the Type I interferon response, we targeted the interferon regulatory factor (IRF) family transcription factor IRF5 that was 3.4-fold upregulated by *Chlamydia* infection. IRFs are a family of transcription factors involved in Type I interferon responses following infection with viruses or intracellular bacteria including *Mycobacterium tuberculosis*[25]. In particular IRF5 has known roles in virus-mediated activation of interferons, as well as modulation of cell growth, regulation of macrophage polarization, apoptosis, and immune system activity[26]. Infection of $IRF5^{-/-}$ human iPSdMs with *C. trachomatis* led to a $45.4 \pm 11.1\%$ increase in bacterial load as compared to the parental KOLF2 iPSdMs (Fig. 7a). Cytokine expression by $IRF5^{-/-}$ human iPSdMs infected with *C. trachomatis* revealed decreased production of the type I interferon IFN-β ($0.35 \pm 0.05$ fold), and pro-inflammatory cytokines, IL-1β ($0.16 \pm 0.03$ fold), IL-6 ($0.24 \pm 0.02$ fold) and TNF-α ($0.14 \pm 0.01$ fold), while production of the anti-inflammatory cytokine, IL-10, was increased ($1.28 \pm 0.03$ fold) compared to the parent KOLF2 iPSdMs (Fig. 7b). This indicated a significant role for IRF5 in macrophages in mediating type I interferon and pro-inflammatory responses to *Chlamydia*, potentially as a mechanism to limit intracellular growth of the bacteria.

To further analyse the impact of knocking out IRF5, we performed RNA-Seq on uninfected and infected macrophages derived from this mutant compared to the parent KOLF2 cells. A total of 1,046 genes were dysregulated when comparing mutant and wild-type uninfected cells, representing 246 upregulated genes and 800 downregulated genes, while comparing mutant and wild-type infected cells the numbers were 995 dysregulated genes comprising 210 upregulated genes and 785 downregulated genes (Supplementary Data 10). Comparing the effect of infection on $IRF5^{-/-}$ iPSdM with that on parent KOLF2 cells, this reflected 21 different pathways (largely downregulated genes), including interferon α/β and γ signalling (adjusted *P* values 8.8E − 162 and 1.7E − 12), negative regulators of RIG1-MDA5 signalling (1.3E − 8), integrin cell surface interactions (7.9E − 41), collagen degradation (3.4E − 35), CRMP proteins in Sema3A signalling and semaphorin interactions (5E − 18 and 5.4E − 9), and L1CAM interactions (1.3E − 6) (Supplementary Data 11). Figure 8 shows a network view of the impact of the $IRF5^{-/-}$ mutation during *Chlamydia* infection with large circles demonstrating genes that

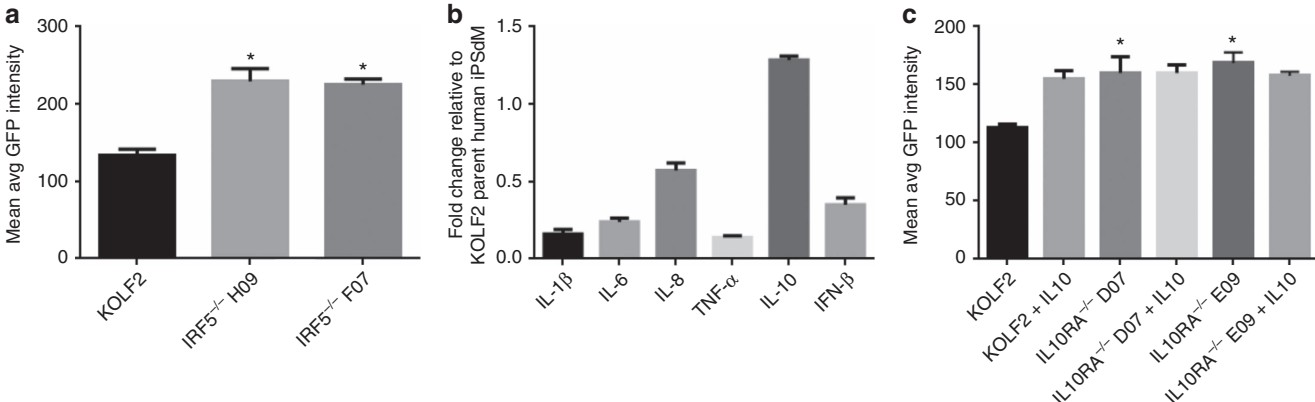

**Figure 7 | *C. trachomatis* infection of human iPSdM CRISPR/Cas9 mutants.** (**a,c**) Level of GFP-tagged *C. trachomatis* infection of human iPSdM CRISPR/Cas9 mutants after 48 h. Results are the averages from three independent measurements ± s.d. using the Cellomics CellInsight NXT. (**b**) Production of cytokines from the *IRF5*$^{-/-}$ mutant. Results are the average from three biological replicates assessed using the Luminex Multiplex and shown as fold change relative to expression of each cytokine in KOLF2 parent iPSdMs. *Represent statistically significant different ($P < 0.05$) between parent and mutants as determined using two-way ANOVA. Equal numbers of WT and mutant cells were seeded for each experiment.

were potentially key hubs, for example, fibronectin (FN), epidermal growth factor receptor, transcriptional regulator SMAD3, DNA replication initiation factor MCM6 and Cyclin-dependent kinase CDK.

RNA-Seq analysis on the dysregulated genes in macrophages after *Chlamydia* infection (Supplementary Data 1), indicated a threefold upregulation of interleukin-10 receptor subunit alpha (IL-10RA) during *C. trachomatis* infection of iPSdMs. IL-10RA is structurally related to the interferon receptors and mediates the immunosuppressive signal of IL-10. IL-10, upregulated 150-fold by *Chlamydia* infection (Fig. 4), is a key immune regulator during infection with various pathogens, including bacteria, viruses and protozoa; evidence here also indicated a prospective regulatory role in *Chlamydia* infection of iPSDMs (Figs 4 and 7c). Human iPSdMs derived from an *IL-10RA*$^{-/-}$ mutant iPS did not respond to IL-10 stimulation (Fig. 7c), consistent with the important role of IL-10RA in mediating IL-10 signalling. Interestingly, the *IL-10RA*$^{-/-}$ mutant iPSDMs were more susceptible to *Chlamydia* infection *in vitro* with a 64.9 ± 12.5% increase in bacterial load compared to parent KOLF2 iPSdMs (Fig. 7c).

## Discussion
Due to the inherent difficulty in genetically manipulating primary monocytes/macrophages, the existing model systems for studying macrophage genetics are largely limited to immortalized cell lines. However, immortalized cells are far removed from the normal genetic state, whereas iPSdMs can be generated from diverse human genetic sources (HipSci project)[27]. We overcame this challenge by using human iPSCs in combination with CRISPR/Cas9 gene editing technology. Here, we reported that macrophages derived from human iPSCs support the entire life cycle of *C. trachomatis*. We show, through a combination of flow cytometry analyses, microscopy and transcriptomic/proteomic profiling, that human iPSC-derived macrophages (iPSdMs) respond to *C. trachomatis* in a similar manner to macrophages derived from human blood monocytes, when both are differentiated in such a way as to yield permissive[7,8] M0/M2-like macrophages. Importantly, the diverse omic-analyses led us to the identification of a number of host factors potentially involved in macrophage–*Chlamydia* interaction.

Macrophages can mediate protection against *Chlamydia* infections by phagocytosis, leading to the destruction of the bacteria. However, some pathogenic *Chlamydia* can infect and avoid lysosomal fusion to survive in macrophages[6]. Here we showed *in vitro* infection of human iPSdM with *C. trachomatis*. Those bacteria that managed to evade lysosomal fusion continued to replicate and developed mature inclusions inside the human iPSdM (Fig. 2). Furthermore, the chlamydial progeny released from infected iPSdM were highly infectious (Fig. 3). Our results thus support the capability of *Chlamydia* to exploit circulating monocytes/macrophages as vehicles for dissemination of infection from the primary infection site to distant sites in the body[28]. Nevertheless our functional genomics studies revealed insights into factors and pathways that may be involved in *Chlamydia* growth in macrophages.

We and others have shown that *Chlamydia* infection induces a Type I interferon (IFN) response in macrophages[29,30]. Activation of the IFN regulatory factors (IRF)s play crucial roles in transcriptional activity of Type I IFN genes and numerous IFN-stimulated genes[31]. The IRF family of transcription factors comprises nine members (IRF1-9). Three of the transcription factors, namely IRF1, 3 and 7 have been characterized as crucial contributors in the regulation of type I IFNs during *Chlamydia* infection[32,33]. Here we explored the contribution of another key member of the IRF family, IRF5, a critical immune regulator. Intriguingly, it has been shown that during viral infection, IRF5 induces overlapping and distinct sets of genes compared to IRF7 (ref. 34). IRF5 not only induces the transcription of IFNα and IFNβ, but also plays a key role in polarizing macrophages towards a pro-inflammatory phenotype and has been associated with susceptibility to inflammatory and autoimmune diseases[35,36]. However, the role of this transcription factor in *Chlamydia* infection has not been investigated to date.

After generation of a human iPS knockout mutant in the IRF5 gene, it was demonstrated that the *IRF5*$^{-/-}$ mutant human iPSdMs showed increased susceptibility to *C. trachomatis* infection compared to the parent KOLF2 human iPSdMs. RNA-Seq analysis of the *IRF5*$^{-/-}$ mutant versus KOLF2 human iPSdMs revealed the dysregulation of a variety of other factors involved in interferon signalling, including STAT1, IRF7 and IFITs. This provides additional evidence that elements of the Type I interferon pathway are essential for anti-chlamydial defence. Overall we can conclude that IRF5 likely controls critical

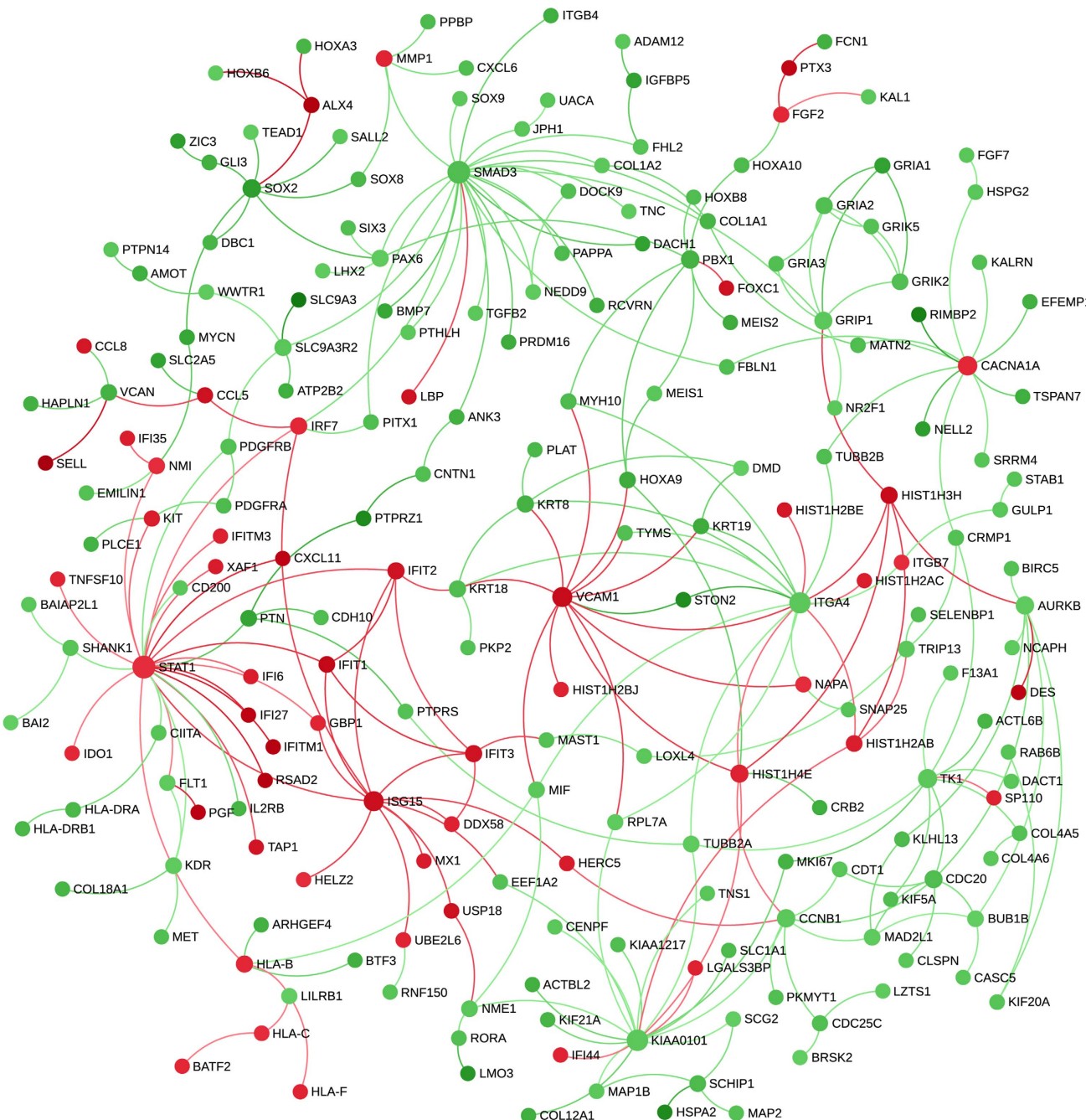

**Figure 8 | Network of expression changes due to infection in the *IRF5*$^{-/-}$ mutant compared to parent KOLF2 cells.** Green = decreased expression; Red = increased expression. RNA-Seq data for differentially expressed genes due to infection (that is, differences between *IRF5*$^{-/-}$ mutant infected/ uninfected compared to parent KOLF2 cells infected/uninfected) were submitted to NetworkAnalyst[45] and interconnected based on known protein:protein interactions (www.innatedb.ca).

mechanisms that limit the excessive growth of *Chlamydia* in macrophages, enabling macrophages more time to exert their role in defence against *Chlamydia*. In this regard, in infected *IRF5*$^{-/-}$ macrophages, we observed substantial downregulation of cytoskeleton components ((FN, the most evident hub, and tubulin (TUBA1)), required for macrophage function, and extracellular matrix, especially collagen, required for extrinsic interactions with other cells and tissues.

IL-10 signals through interaction with its corresponding receptor, IL-10R. IL-10R consists of two chains, α and β. IL-10/ IL-10RA/B interaction leads to the activation of several intracellular signalling pathways resulting in pleiotropic effects,

including the suppression of inflammation and induction of apoptosis. Associations of genetic variations (polymorphisms) in IL-10 with *Chlamydia* disease pathogenesis have been reported[37]. The role of IL-10/IL-10R signalling in *Chlamydia* infection is likely to be complex. While studies have shown that IL-10 supressed expression of several pro-inflammatory molecules that might be involved in immune responses against *Chlamydia* infections[38], we and others[39] have shown that *Chlamydia* also induces the expression of IL-10 and IL-10R. In this study we report that blocking IL-10 signalling by a homozygous knockout of IL-10RA in human iPSdMs led to increased susceptibility to *C. trachomatis* infection. This supports a role for IL-10 in limiting

*Chlamydia* infections of macrophages. Chlamydiae have evolved a number of strategies to promote their survival inside host cells, one of which is modulation of programmed cell death pathways in infected host cells[40]. A previous study showed that *Chlamydia* inhibits apoptosis in human peripheral blood mononuclear cells through induction of IL-10 (ref. 41). Therefore, we speculate that the absence of IL-10/IL-10R signalling might inhibit apoptosis in macrophages to promote growth of *Chlamydia* in these cells. We are currently investigating this mechanism experimentally.

In conclusion, we have provided evidence that we can use the human iPSdM *Chlamydia* infection model to study the complex interplay beween the host and the pathogen.

## Methods

**Ethical approval.** All experiments were approved by and carried out in accordance with the Wellcome Trust Sanger Institute and UBC Research Ethics Board approved guidelines.

**Human iPSC culture and differentiation to macrophages.** The human iPS lines used in this study, KOLF2 and FPS10C, were generated by the Sanger Institute's Human Induced Pluripotent Stem Cells Initiative (HipSci) project. Briefly, the HipSci Initiative used the CytoTune 1 Sendai method to reprogram dermal fibroblast obtained from healthy adult volunteers into iPSCs. The Certificate of Analysis for the human iPSCs and more detailed methods on the generation of iPSCs are available on the project website (www.hipsci.org). Undifferentiated human iPSCs were grown under feeder-free TeSR-E8 conditions (StemCell Technologies) on tissue culture plates coated with Synthemax II-SC substrate (Corning) as per manufacturer's instructions. Interestingly it has been shown that *in vitro* human embryonic stem cell hematopoiesis mimics MYB-independent yolk sac hematopoiesis[42]. We compared iPSdMs with human blood-derived macrophages since the latter are derived in an analogous manner from progenitor cells and represent the most common model for studying *ex vivo* macrophage biology. To differentiate human iPSCs to macrophages, we adapted and modified the approach of Hale *et al.*[13] (Fig. 1a). Briefly, upon reaching confluency, the human iPSCs were collected and transferred into iPS base medium (DMEM/F12 supplemented with 20% KnockOut Serum Replacement, 2 mM L-Glutamine, 0.055 mM β-mercaptoethanol) with 10 ng ml$^{-1}$ BMP-4 (R&D) for 4 days to generate Embryoid Bodies (EBs). On Day 5, EBs were used for generation of myeloid precursor cells in the presence of 25 ng/ml IL-3 (R&D) and 50 ng ml$^{-1}$ M-CSF (R&D) followed by terminal differentiation and maturation of myeloid precursors into matured macrophages (iPSdMs) in the presence of higher concentrations of M-CSF (100 ng ml$^{-1}$). For experiments, macrophages were detached using Lidocaine solution (4 mg ml$^{-1}$ lidocaine-HCl with 10 mM EDTA in PBS), and seeded at $2 \times 10^5$ cells per well (24-well plate) or $1 \times 10^6$ cells per well (six-well plate).

***C. trachomatis* propagation and infection of macrophages.** *C. trachomatis* strains Sweden F plasmid minus (SWFP − ) and SWFP − transformed with plasmid pGFP::SW2 used in this study were kindly provided by Ian Clarke[15], propagated in McCoy fibroblasts (ATCC) and purified using a Gastrografin step gradient, as described previously[43]. Purified EBs were stored in sucrose/phosphate/glutamate buffer in aliquots at − 80 °C until use.

Macrophages were seeded into 24-well plates in RPMI 1640 media (Sigma)-supplemented with 10% heat-inactivated FBS (Sigma) and 2 mM L-Glutamine (Sigma) and allowed to settle overnight. On the day of infection, cells were washed and replaced with fresh RPMI-supplemented media. *C. trachomatis* were added to the macrophages at MOI of 1 and the plates were centrifuged at 1,800 r.p.m. for 30 min at 37 °C. Cells were washed, replaced with fresh media and incubated at 37 °C and 5% $CO_2$ for the indicated time point. At the time point, supernatants were harvested, filtered and stored in − 80 °C for cytokine/chemokine analysis. Cells were washed, fixed with 4% paraformaldehyde (PFA) and stained with Hoechst 33342 (1:1,000; Thermo Fisher Scientific). The average intensity of GFP expressed by internalized *C. trachomatis* was measured using the Target Activation BioApplication in a CellInsight NXT High Content Screening (HCS) Platform (Thermo Fisher Scientific, Pittsburgh, PA). All HCS experiments were performed at resolution of × 10 magnification. Alternatively for electron microscopy, cells were fixed with a mixture of 2.5% glutaraldehyde and 4% PFA on ice for 1 h and processed as described[44]. As indicated, macrophages were also treated with a competitive inhibitor of IDO1 (INCB024360, Selleckchem) at 500 nmol for 24 h prior to *C. trachomatis* infection.

**Cytokine/chemokine multiplex bead assays.** Supernatants from *C. trachomatis* infected macrophages were analysed for production of human tumour necrosis factor α (TNF-α), interleukin 1β (IL-1β), 6 (IL-6), 8 (IL-8), 10 (IL-10) and interferon β (IFN-β). TNF-α, IL-1β, IL-6, IL-8 and IL-10 were measured using a Millipore customised Milliplex anti-human cytokine kit and data acquired

on a Luminex FLEXMAP 3D system (Millipore). Data analysis was carried out using the xPONENT Multiplex Assay Analysis software (Millipore). Production of IFN-β was measured using the human IFN-β ELISA kit (Thermo Fisher Scientific).

**Flow cytometry.** Macrophages were lifted off from the surface of culture wells, fixed with 1% PFA and stained with specific conjugated antibodies, including CD11b (FITC mouse anti-human CD11b, BD Pharmingen catalog no. 562793, 5 µl per $10^6$ cells), CD14 (APC mouse anti-human CD14, BD Pharmingen catalog no. 561383, 5 µl per $10^6$ cells), CD16 (Pacific Blue mouse anti-human CD16, BD Pharmingen catalog no. 558122, 0.2 mg ml$^{-1}$), CD68 (Alexa Flour 647 mouse anti-human CD68, BD Pharmingen catalog no. 562111, 5 µl per $10^6$ cells), SSEA-4 (Alexa Fluor 488 mouse anti-SSEA-4, BD Pharmingen catalog no. 560308, 5 µl per $10^6$ cells) and OCT-3/4 (PE mouse anti-Oct3/4, BD Pharmingen catalog no. 561556, 5 µl per $10^6$ cells). For intracellular markers, cells were permeabilized with saponin buffer. Isotype matched antibodies conjugated to the same fluorophores were used as negative controls. Samples were analysed using the BD LSRFortessa (BD Biosciences). Data were analysed using FlowJo v10.1.

**RNA and protein isolation for RNA-Seq and Proteomics.** Gene expression was measured by RNA-Seq at 1 and 24 h post infection and protein expression was measured by Proteomic Mass Spectrometry at 24 h post infection. Briefly, total RNA and proteins from three independent samples of uninfected and *C. trachomatis* infected human iPSdMs at the indicated time points were isolated using the AllPrep Mini Kit (Qiagen).

For RNA-Seq, the Illumina TruSeq RNA Sample Preparation v2 Kit was used and poly-A tailed RNA (mRNA) was purified via oligo dT magnetic bead pull down. The mRNA was than fragmented using metal ion-catalysed hydrolysis. A random-primed cDNA library was synthesized and the resulting double-stranded cDNA was used as the input for library preparation. Overhangs were repaired with a combination of fill-in reactions and exonuclease activity to produce blunt ends. Adenylation of blunt ends was followed by ligation to Illumina Pair-end Sequencing adapters containing unique index sequences. cDNA enrichment was carried out by 10 cycles of PCR amplification. Samples were quantified and pooled based on post-PCR analyses with an Agilent Bioanalyzer. Pools were size-selected using the LabChip XT Caliper. The multiplexed library was then sequenced on the Illumina HiSeq 2000, 75 bp paired-end read length.

Sequenced data was then analysed and quality controlled (QC and individual indexed library BAM files were produced). Tophat2 version 2.1.1 was used to align the reads to the reference human genome GRCh37 and indexed with Samtools version 0.1.19. Read counts were computed from the read alignments to the coding sequence using HTSeq version 0.6.0. Differentially expressed genes were calculated by the DESeq2 version 1.12. 3 and R version 3.3.1. Pathway analysis was performed using SIGORA version 2.0.1, which identified pathway enrichment based on statistically over-represented Pathway Gene-Pair Signatures[16]. Network analysis of expression changes were performed using NetworkAnalyst[45] which is based on curated protein:protein interactions from InnateDB (www.innatedb.ca).

For proteomics, protein pellets were dissolved in 0.1 M triethylammonium bicarbonate, 0.1% SDS. Total proteins were reduced with tris-2-carboxymethyl phosphine and cysteine residues were blocked with iodacetamide. Samples were digested with Trypsin (Pierce, MS grade) and the resultant peptides were labelled with TMT6plex (Thermo Scientific), and pooled. Offline peptide fractionation based on high pH Reverse Phase (RP) chromatography was performed using the Waters, XBridge C18 column on a Dionex Ultimate 3000 HPLC system equipped with autosampler. Signal was recorded at 280 nm and fractions were collected in a time dependent manner. LC-MS analysis was performed on the Dionex Ultimate 3000 UHPLC system coupled with the Orbitrap Fusion Tribrid Mass Spectrometer (Thermo Scientific). Each peptide fraction was reconstituted in 0.1% formic acid and loaded to the Acclaim PepMap 100 and subjected to a multi-step gradient elution on the Acclaim PepMap RSLC C18 capillary column (Dionex) retrofitted to an electrospray emitter (New Objective, FS360-20-10-D-20). Precursors between 400 and 1,500 $m/z$ were selected with mass resolution of 120 K, AGC $3 \times 10^5$ and IT 100 ms were isolated for CID fragmentation with quadrupole isolation width 0.7 Th. MS3 quantification spectra were acquired with further HCD fragmentation of the top 10 most abundant CID fragments isolated with Synchronous Precursor Selection (SPS) excluding neutral losses of maximum $m/z$ 30. The HCD MS3 spectra were acquired with resolution of 15 K. Targeted precursors were dynamically excluded for further isolation and activation for 45 s with 7 p.p.m. mass tolerance.

The acquired mass spectra were submitted to SequestHT search engine implemented on the Proteome Discoverer 1.4 software for protein identification and quantification. The precursor mass tolerance was set at 20 p.p.m. and the fragment ion mass tolerance was set at 0.5 Da. Spectra were searched for fully tryptic peptides with a maximum of two mis-cleavages and a minimum length of six amino acids. TMT6plex at N-terminus, K and Carbamidomethyl at C were defined as static modifications. Dynamic modifications included oxidation of M and deamidation of N, Q. A maximum of two different dynamic modifications were allowed for each peptide with maximum two repetitions each. Peptide confidence was estimated with the Percolator node. Peptide FDR was set at 0.01 and validation was based on $q$-value and decoy database search. All spectra were searched against a UniProt fasta file containing 20 K human reviewed entries.

The Reporter Ion Quantifier node included a custom TMT 6plex Quantification Method with integration window tolerance 20 p.p.m. and integration method the Most Confident Centroid at the MS3 level.

**Generation of biallelic knockouts in human iPSCs.** Due to limitations in our ethics certification that did not allow for genetic manipulations in the FPS10C iPSC line, only the KOLF2 human iPSCs was utilized for mutant generation. KOLF2 cells were adapted to feeder-free culture and several sublines were isolated by single cell cloning. The subline KOLF2-C1 was used for this study and these cells showed a stable, normal karyotype (46;XY) for up to 25 passages. Biallelic knockouts in KOLF2 human iPSCs were generated using a method that was found to minimize the potential for off-target effects[12]. Briefly, the intermediate targeting vector for each gene was generated by GIBSON assembly of the four fragments: puc19 vector, 5′ homology arm, R1-pheS/zeo-R2 cassette and 3′ homology arm. The homology arms were amplified by PCR from KOLF2 human iPSC genomic DNA (PCR primer sequences are listed in Supplementary Table 1). pUC19 vector and R1-pheS/zeo-R2 cassette were prepared as gel purified blunt fragments (EcoRV digested) while the PCR fragments were either gel purified or column purified (QIAquick, QIAGEN). The resultant GIBSON assembly reactions (Gibson Assembly Master Mix, NEB) were transformed into NEB 5-alpha competent cells and clones resistant to carbenicillin ($50\,\mu g\,ml^{-1}$) and zeocin ($10\,\mu g\,ml^{-1}$) were analysed by Sanger sequencing to verify all junctions (sequencing primers are listed in Supplementary Table 1). Subsequently, the intermediate targeting vectors were turned into donor plasmids via a Gateway exchange reaction. LR Clonase II Plus enzyme mix (Invitrogen) was used as described by Skarnes et al.[46] with the difference that it was a two-way reaction exchanging only the R1-pheSzeo-R2 cassette with the pL1-EF1αPuro-L2 cassette. The latter had been generated by cloning synthetic DNA fragments of the EF1α promoter and puromycin into one of the pL1/L2 vectors as described by Skarnes et al.[46]. Following Gateway reaction and selection on YEG + carbenicillin ($50\,\mu g\,ml^{-1}$) agar plates, correct donor plasmids were confirmed by Sanger sequencing of all junctions. Plasmids carrying single guide (sg) RNA sequences were generated by cloning forward and reverse strand oligos into the BsaI site of either U6_BsaI_gRNA or p1260_T7_BsaI_gRNA vectors (kindly provided by Sebastian Gerety, unpublished) (CRISPR sequences are listed in Supplementary Table 1). Kanamycin resistant clones ($50\,\mu g\,ml^{-1}$) were isolated and cloning of the correct sequence was verified by Sanger sequencing.

Human iPSCs were dissociated to single cells and nucleofected (Amaxa2b nucleofector, LONZA) with Cas9 coding plasmid (hCas9, Addgene 41815), sgRNA plasmid and donor plasmid. Following nucleofection, cells were selected for up to 11 days with $0.25\,\mu g\,ml^{-1}$ puromycin. Individual colonies were picked into 96-well plates, grown to confluence and then replica plated. Once confluent, the replica plates were either frozen as single cells in 96-well vials or the wells were lysed for genotyping.

To genotype individual clones from a 96-well replica plates, cells were lysed and used for PCR amplification with LongAmp Taq DNA Polymerase (NEB). Insertion of the cassette into the correct locus was confirmed by visualising on 1% E-gel (Life Tech) PCR products generated by gene specific (GF1 and GR1) and cassette specific (ER and PF) primers for both 5′ and 3′ ends (Supplementary Table 1). We also confirmed single integration of the cassette by performing a qPCR copy number assay. To check the CRISPR site on the non-targeted allele PCR products were generated either from across the locus, using the 5′ and the 3′ gene specific genotyping primers (GF1-GR1), or from around the site using primers 5F-3R (Supplementary Table 1) that would amplify a short, around 500 bp, amplicon. In both cases the PCR products were treated with exonuclease and alkaline phosphatase (NEB) and Sanger sequenced using primers SF and SR (Supplementary Table 1). Sequence reads and their traces were analysed and visualized using a laboratory information management system (LIMS)-2. To minimize the potential for off-target effects, two independently derived clones with different specific mutations were isolated for each of the targeted genes and studied further here.

**Statistical analyses.** Statistical significance was performed with GraphPad Prism software. Student's t-test was used for two-group comparisons, and ANOVA was used for comparisons involving three or more groups.

**Data availability.** The RNA-seq data have been deposited in the European Nucleotide Archive with accession code ERP006216. The mass spectrometry proteomics data have been submitted to the ProteomXchange Consortium via the PRIDE partner respository with the data set identifier PXD005858. The authors declare that all other relevant data supporting the findings of this study are included in this published article and its Supplementary Information files, or from the corresponding author upon request.

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

## Acknowledgements

This work was supported by The Wellcome Trust. REWH would like to acknowledge funding from the Canadian Institutes for Health Research (CIHR, funding reference number MOP-363437) for the work reported here and personal support from the Canada Research Chair program and University of British Columbia Killam Professorship. We thank Dr Ian Clarke for providing us with the *C. trachomatis* strains Sweden F plasmid minus (SWFP − ) and SWFP − transformed with plasmid pGFP::SW2. We thank Theresa Feltwell, Louise Ellison and Dr Helena Smith for sharing *Chlamydia* propagation protocols. We thank Drs Rey Carabeo and Tristan Thwaites for teaching us how to perform density gradient purification of *Chlamydia*. We thank Gareth Griffiths and Katie Andrews for help with genotyping.

## Author contributions

Conceived and designed experiments: A.T.Y.Y., C.H., G.D. Performed the experiments: A.T.Y.Y., D.G., T.R. Analysed the data: A.T.Y.Y., A.H.L., E.E.G., T.R., J.C., R.E.W.H. Contributed reagents/material/analysis tools: W.B., D.P.-S., D.P., N.T., W.C.S., R.E.W.H. Wrote the paper: A.T.Y.Y., A.H.L., T.R., W.C.S., R.E.W.H., G.D.

## Additional information

**Competing interests:** The authors declare no competing financial interests.

