## [Peer Review File · Nature Communications]

Reviewers' comments:

Reviewer #1 (Remarks to the Author):

This is an interesting study on the usability of iPSC to establish human infection models for *Chlamydia trachomatis* infections. The authors derived macrophages from iPSCs and demonstrated the infection of these cells with *Chlamydia*. Transcriptome and proteome analyses were performed to catalogue the changes induced by *Chlamydia* infection. They further developed a CRISPR/Cas9-based approach to genetically manipulate these cells and demonstrated the impact of IRF5 and IL-10RA in controlling *Chlamydia* infection in this model.

Establishment of authentic human infection models for the investigation of human pathogens is very important to be able to draw conclusions on the mechanisms underlying pathogenicity. This study demonstrates the enormous potential of iPSC as a source to generate infection models for *Chlamydia*. The question is whether the macrophages generated from iPSC really have the features of human macrophages as has been stated by the authors.

Since this is the first report on the use of iPSCs the authors should undertake more effort to compare them with human macrophages. They did this nicely for the transcription response to *Chlamydia* infection but further details on the infection phenotypes are required. *C. trachomatis* does not replicate efficiently in human macrophages. At least many serovars and strains tested so far failed to produce significant progeny and form large inclusions in primary macrophages. Since this is apparently entirely different in iPSCs (see figure 2B), the question is whether iPSCs really represent infection models comparable to primary macrophages. The authors should compare and quantify the infection efficiency and progeny formation of the *Chlamydia* strain used in the study grown in primary macrophages and iPSCs.

There is evidence in the literature that CRISPR/Cas9 generates off-target mutations in human embryos. Since the authors established a new strategy for gene knockout here it is important to evaluate the possibility of off-target effects. This could be done by testing a second clone with a different mutation in the genes of interest or by sequencing.

Supplementary figure 1 indicates that the mutation in the knockout cell lines were validated by PCR, sequencing and mass spectrometry. It is not clear what is shown in figure S1. What does PF/GR1 and GF1/ER mean? What is the conclusion from the data shown in figure S1? Where are the mass spectrometry data? It is important to show a loss of protein expression either by mass spec or immunoblotting in the knockout clones.

Figure 7 requires statistical analysis.

Reviewer #2 (Remarks to the Author):

In the manuscript 'Exploiting induced pluripotent stem cell-derived macrophages to unravel key host factors influencing *Chlamydia trachomatis* pathogenesis' the authors aim to overcome the limitations in the availability of (relevant) in vitro models in the study of complex macrophage response of the bacterial pathogen *Chlamydia trachomatis*. They have done this using human induced pluripotent stem cells (iPSCs) to study macrophage-*Chlamydia* interactions in vitro and using transcriptomic and proteomic profiling. They have further generated biallelic knockout mutations in iPSCs based on CRISPR/Cas9 to identify human host genes significantly influencing *Chlamydia* pathogenesis.

***In particular, we would be grateful for your comments on the transcriptomic and proteomic

analyses included in the report.***

I have only commented on methods and results related to the 'omics' analysis in the work.

The authors have used the conventional Tophat-samtools-HTSeq pipeline followed by the R module 'DeSeq2' for identifying differentially expressed genes using RNASeq technique. This is known to work well and produce acceptable results. The RNASeq data points were sampled at 1 hr and 24 hrs post infection. Since there is no clear mention, I am assuming the analysis was done with at least three replicates (or more) as is the accepted norm in transcriptomics analysis. It will be good to mention the details in the manuscript.

A total of 2,029 genes repeatedly changed expression after Chlamydia infection in both iPScMs and blood derived macrophages, with 1,194 genes upregulated and 835 gene downregulated (fold change of ≥ 1.5 , $p \leq 0.05$; see Supplementary Table 2 for list of selected DE genes;

Does the above sentence mean that there were more genes specific to the two experiments, which were found differentially expressed, in addition to the 2029 genes found changing in both iPScMs and blood derived macrophages? If so it would be prudent to discuss them. Were the non-common DE genes a big fraction of the total set of differentially expressed genes? If so, can the author comment more on this?

The use of SIGORA pathway tool for identifying over-represented Pathway Gene-Pair signatures to avoid repetitive assignment of the same genes to multiple overlapping pathways makes sense. Overrepresentation of immune related pathways in infected cells is expected for an infection assay and the subsequent inferences based on 'Pathway Gene-Pair signatures' make sense. The extremely high fold change for the IDO1 gene related to defense mechanisms as identified by RNASeq in infected human iPScMs is interesting. The interpretation of the 835 down-regulated genes is rather hypothetical and needs further work.

The proteomics pipeline and the various parameter cutoffs used for the analysis makes sense. Considering the limited sensitivity encountered in proteomics, it is good to see the overlap of the interferon related and immune pathways between proteomics and transcriptomics.

Overall, apart from a few details (regarding the number of replicates) and the significance of the non-common differentially expressed genes, I find the approach and the results presented in the transcriptomics and proteomic analysis satisfactory.

Reviewer #3 (Remarks to the Author):

The paper by Yeung et al., demonstrates the advantages of using iPSC technologies for the study of Chlamydia pathogenesis. The authors showed that macrophages derived from iPSCs (iPScMs) could be infected with Chlamydia. In addition, they genetically modified iPSCs using CRISPR/Cas9 to knockout IRF5 and IL10RA and found that lack of these genes increased susceptibility of iPScMs to Chlamydia infection. RNAseq and proteomic analysis has been performed to determine how iPScMs respond to Chlamydia.

Overall, paper demonstrated the value and advantages of iPSC technologies for study of Chlamydia pathogenesis. However, it would be nice to see how the response of iPScM to Chlamydia infection is different from somatic macrophages.

During development, macrophages arise from different waves of hematopoiesis. In mouse system Myb-independent and Myb-dependent waves of embryonic hematopoiesis were identified. Authors

should comment whether their differentiation protocol recapitulate Myb-dependent or Myb-independent macrophage pathway.

Does knockout of IRF5 and IL10R affect the yield of macrophages?

Supplementary Figure 3d should include data from wild type to allow comparison wild and knockout iPSCs.

Materials and methods should provide references describing iPSC lines used in studies or indicate the method used for iPSC- generation (lenti, episomal, Sendai?).

Reviewer #4 (Remarks to the Author):

Yeung et al. describe a system for editing iPSCs using CRISPR, then differentiating into macrophages, which are used as an in vitro model for studying Chlamydia infections. As the authors have pointed out, the use of iPSC-derived, genetically modified macrophages may prove to be a useful system for studying host-pathogen interactions in general, and in the case of studying Chlamydia infections this approach appears to be novel.

As far as method development goes, the editing of iPSC cells with CRISPR – including the use of selection markers to increase efficiency of editing – is fairly well established. The authors' claim of having "not only improved the frequency of biallelic mutations, but also greatly simplified the final genotyping step of the mutant clones by requiring only Sanger sequencing, and is thus particularly useful for the generation of biallelic mutants at scale" is not all at substantiated by the data presented here, and in fact an irresponsible claim to make. The only data presented here, in Figure 6 and Supplementary Data Figure 1 give no indications of the general efficiencies of editing the iPSCs with or without the selection marker, or number of sub-clones assayed and their genotypes. There are several big problems here:

- The PCR assay in Supp Fig 1, its unsuitability for genotyping aside, is essentially uninterpretable: what are the lanes? Where are the controls?
- The use of Sanger sequencing for genotyping is also problematic. It's fundamentally a population-level assay, and easily masks any heterogeneity in not truly clonal populations.
- In the IRF5 clone, is the 96bp deletion in frame? And if so does it truly abolish the activity of IRF5?
- Have the authors looked at whether the selection marker has integrated into other sites in the genome, or whether Cas9 editing may have affected the expression of any other gene (i.e. all kinds of possible off-target effects)? Without having done a careful genotyping analysis it's really not appropriate to be drawing any functional conclusions.

RESPONSE TO REVIEWERS' COMMENTS:

Reviewers comments are provided in italics with our responses in plain text.

REVIEWER #1 (*Remarks to the Author*):

This is an interesting study on the usability of iPSC to establish human infection models for Chlamydia trachomatis infections. The authors derived macrophages from iPSCs and demonstrated the infection of these cells with Chlamydia. Transcriptome and proteome analyses were performed to catalogue the changes induced by Chlamydia infection. They further developed a CRISPR/Cas9-based approach to genetically manipulate these cells and demonstrated the impact of IRF5 and IL-10RA in controlling Chlamydia infection in this model.

*Establishment of authentic human infection models for the investigation of human pathogens is very important to be able to draw conclusions on the mechanisms underlying pathogenicity. This study demonstrates the enormous potential of iPSC as a source to generate infection models for Chlamydia. **The question is whether the macrophages generated from iPSC really have the features of human macrophages as has been stated by the authors.***

Since this is the first report on the use of iPScDMs the authors should undertake more effort to compare them with human macrophages. They did this nicely for the transcription response to Chlamydia infection but further details on the infection phenotypes are required. C. trachomatis does not replicate efficiently in human macrophages. At least many serovars and strains tested so far failed to produce significant progeny and form large inclusions in primary macrophages. Since this is apparently entirely different in iPScDMs (see figure 2B), the question is whether iPScDMs really represent infection models comparable to primary macrophages. The authors should compare and quantify the infection efficiency and progeny formation of the Chlamydia strain used in the study grown in primary macrophages and iPScDMs.

We thank the reviewer for his comments and in particular his recognition of the “enormous potential of this model”. Over the past decade, there have been several publications documenting the use of human iPSCs as a preferable *in vitro* modelling system to study host-pathogen interactions [Hale C. et al. 2015. PLoS One 10:e0124307; Trevisan M et al. 2015 Viruses 7:3835-56]. In this study we have adapted this in a novel fashion by adapting the model to an obligate intracellular pathogen, and further shown that human genetic methods can be applied to this system to understand the genetics of infection. The method used in this study to differentiate human iPSCs into macrophages has been previously published [Hale C. et al. 2015. PLoS One 10:e0124307]. The resultant iPSC differentiated macrophages (iPScDMs) have been extensively compared with macrophages derived from

human blood monocytes [Alasoo K. et al. 2015. Sci Rep 5:12524] and this is further extended here. Detailed transcriptomic and phenotypic results from two published papers have shown that iPScDMs are comparable to macrophages derived from human blood monocytes and both papers came to the conclusion that iPScDMs serve as good *in vitro* models for infectious diseases modelling [Hale C. et al. 2015 PLoS One 10:e0124307; Alasoo K. et al. 2015 Sci Rep 5:12524]. In this study we differentiated human iPSCs into macrophages and used electron microscopy to show our iPScDMs morphologically resembled macrophages that were similarly derived from human blood monocytes. We also showed that iPScDMs expressed markers specific to macrophages and showed deep overlaps in unstimulated transcriptional profiles and responses to *Chlamydia*.

Using such a tractable system has allowed deep genomic insights into the responses of iPScDMs to *Chlamydia trachomatis*. As *C. trachomatis* preferentially infects epithelial cells, the molecular details of how *Chlamydia* infect their host cells have been best investigated in epithelial cell lines that are difficult to genetically manipulate. Conversely the molecular details of how *Chlamydia* interact with macrophages are less well studied but we submit that this cell type is critical in infection. Clinically, *C. trachomatis* have been detected in the synovial tissues of patients with reactive arthritis [Beutler AM et al. 1995 Am J Med Sci 310:206-13; Gerard HC et al. 1998 J Rheumatol 25:734-43; Nanagara R et al. 1995 Arthritis Rheum 38:1410-7]. It has been further documented that *C. trachomatis* reside predominantly within macrophages below synovial lining and the residing *C. trachomatis* are viable and metabolically active. This argues against the reviewer's comment that *C. trachomatis* cannot infect and replicate within human macrophages. Moreover, it has been documented that the capacity to infect macrophages varies between Chlamydial strains, the source of macrophages and the MOI used for infection [Herweg J-A and Rudel T. FEBS J. 2016 283:608-18]. Indeed recent papers have also shown that the ability of *Chlamydia* to replicate inside macrophages depends on the macrophage sub-type since *Chlamydia* is certainly restricted in inflammatory M1 macrophages but actually grows well in M2 and M0 (resting) macrophages [Gracey E. et al. 2013 PLoS One. 8:e69421; Bushacher T. et al. 2015 PLoS One. 10:e0143593]. Our macrophages appear to be analogous to M0 or resting macrophages and we have now added the above information to the manuscript at lines 38-44, 104-6 and 435-6. In the submitted manuscript we clearly showed very similar replication and gene expression in human blood derived macrophages (which were derived in such a way as to avoid the inhibitory M1 profile) and iPSC derived macrophages.

In this study, we followed the infection of *C. trachomatis* serovar F in iPScDMs using TEM and fluorescence imaging and showed these images in Figure 2. We also performed infections of macrophages derived from human blood monocytes with *C. trachomatis* serovar F at the same MOI, and observed very similar phenotypes. We have now added images of *Chlamydia* infection progression in both human blood monocyte-derived macrophages and human iPScDMs for comparison in Figure 2.

There is evidence in the literature that CRISPR/Cas9 generates off-target mutations in human embryos. Since the authors established a new strategy for gene knockout here it is important to evaluate the possibility of off-target effects. This could be done by testing a second clone with a different mutation in the genes of interest or by sequencing.

At Sanger, we have performed extensive method development to ensure that we did not have off-target effects. Indeed, our stem cell engineering (Allan Bradley, Bill Skarnes) teams have now made multiple CRISPR mutants at Sanger in both mouse and human lines and are regularly monitoring off target impact. The targeting strategy we had used for this paper has been validated in a recently accepted paper in Development from our co-author W. Skarnes with S. Pollard [Bressan R.B. et al. 2017 Development. in press], where they generated

knockout mutations in more than 10 transcription factors and did not observe off-target effects in any of the knockout mutant generated. However to overcome this valid critique we have now added characterization of a second independently-derived mutant for each of the two described knockouts; these would be anticipated to have completely different off-target effects but have almost identical impacts in our *Chlamydia* infection studies. We have now included in Figure 7 *Chlamydia* infection data for two clones with different mutations in IRF5 and IL10RA. Both clones for each target gene showed similar levels of increased susceptibility to *C. trachomatis* infection. These issues are now discussed in the text at lines 206-8, 248-250 and 373.

Supplementary figure 1 indicates that the mutation in the knockout cell lines were validated by PCR, sequencing and mass spectrometry. It is not clear what is shown in figure S1. What does PF/GR1 and GF1/ER mean? What is the conclusion form the data shown in figure S1? Where are the mass spectrometry data? It is important to show a loss of protein expression either by mass spec or immunoblotting in the knockout clones.

We apologize that this Supplementary Figure contains mistakes, pointed out by the reviewer, that were overlooked. First of all, we did not present mass spectrometry data so we should have removed the words ‘mass spectrometry’ from the Figure description. Second, we made a mistake that in Fig. S1b IRF5, should have indicated a 97bp deletion, leading to a frameshift, rather than 96bp deletion. We have now added a second frameshift IRF5^{-/-} clone with an insertion of 2 bp in exon 3 (Sequencing data below). Third, Fig. S1a has been amended to include a schematic of where the specific PCR primers PF/GR1 and GF1/ER bind.

Figure 7 requires statistical analysis.

We re-generated the graph to include statistical analysis using Graphpad.

REVIEWER #2 (Remarks to the Author):

In the manuscript ‘Exploiting induced pluripotent stem cell-derived macrophages to unravel key host factors influencing Chlamydia trachomatis pathogenesis’ the authors aim to overcome the limitations in the availability of (relevant) in vitro models in the study of complex macrophage response of the bacterial pathogen Chlamydia trachomatis. They have done this using human induced pluripotent stem cells (iPSdMs) to study macrophage-Chlamydia interactions in vitro and using transcriptomic and proteomic profiling. They have further generated biallelic knockout mutations in iPSCs based on CRISPER/Cas9 to identify human host genes significantly influencing Chlamydia pathogenesis.

****In particular, we would be grateful for your comments on the transcriptomic and proteomic analyses included in the report.****

I have only commented on methods and results related to the ‘omics’ analysis in the work.

The authors have used the conventional Tophat-samtools-HTSeq pipeline followed by the R module ‘DeSeq2’ for identifying differentially expressed genes using RNASeq technique. This is known to work well and produce acceptable results. The RNASeq data points were sampled at 1 hr and 24 hrs post infection. Since there is no clear mention, I am assuming the analysis was done with at least three replicates (or more) as is the accepted norm in transcriptomics analysis. It will be good to mention the details in the manuscript.

Yes, the transcriptomic analysis of the iPSdMs was performed with three biological replicates. This is now mentioned in the methods at line 143.

“A total of 2,029 genes repeatedly changed expression after Chlamydia infection in both iPSdMs and blood derived macrophages, with 1,194 genes upregulated and 835 gene

downregulated (fold change of ≥ 1.5 , $p \leq 0.05$; see Supplementary Table 2 for list of selected DE genes;” Does the above sentence mean that there were more genes specific to the two experiments, which were found differentially expressed, in addition to the 2029 genes found changing in both iPSdMs and blood derived macrophages? If so it would be prudent to discuss them. Were the non-common DE genes a big fraction of the total set of differentially expressed genes? If so, can the author comment more on this?

We did observe differentially expressed genes unique to the iPSdMs and blood monocyte-derived macrophages (MDMs) which were consistent with previous studies on such cells. We have included the additional data in the updated Supplementary Table 3D-3G for the iPSdMs (KOLF2) and MDMs showing unique up- and down- regulated pathways. In KOLF2-unique genes, Sigora enrichment analysis identified a number of pathways including extracellular matrix organization, collagen degradation, cell surface interactions at the vascular wall. In contrast, in MDMs, we observed a number of cellular metabolic processes, including mitochondrial protein import, pyruvate metabolism, gluconeogenesis, as well as antigen processing and cell cycle regulation (G2/M transition and DNA repair). These differences were consistent with previous report that highly expressed genes in MDMs were significantly enriched for antigen binding pathways [Alasoo K. et al. 2015. Sci Rep. 5:12524]. Similarly, genes that were more highly expressed in iPSdMs showed enrichment for cell adhesion and extracellular matrix pathways [Alasoo K. et al. 2015. Sci Rep. 5:12524]. While the non-common DE genes represent a substantial fraction of the total set of differentially expressed genes, we observed that there was a strong over-representation in infected cells of major immune-related pathways in commonly dysregulated genes of iPSdMs and blood macrophages, indicating extensive common responses to infection in these two cell types. This is now mentioned in the text on lines 339-347 and pointed out in the Legend for this Table.

*The use of SIGORA pathway tool for identifying over-represented Pathway Gene-Pair signatures to avoid repetitive assignment of the same genes to multiple overlapping pathways makes sense. Overrepresentation of immune related pathways in infected cells is expected for an infection assay and the subsequent inferences based on ‘Pathway Gene-Pair signatures’ make sense. The extremely high fold change for the IDO1 gene related to defense mechanisms as identified by RNASeq in infected human iPSdMs is interesting. **The interpretation of the 835 down-regulated genes is rather hypothetical and needs further work.***

Of the 835 down-regulated genes commonly observed human iPSdMs and MDMs, we found pathway enrichments in gap junction trafficking and regulation, as well as various translation processes and nonsense mediated decay (NMD) regulation (Supplementary Table S3C). Consistent with previous work [Humphrys MS et al. 2013 PLoS One 8(12):e80597], we observed the down-regulation of gap junction regulation. In eukaryotic cells, post-transcriptional regulation by modulating mRNA translation and mRNA stability is a powerful mechanism to control gene expression in response to changing environmental stimuli. While a number of viral and bacterial effectors have been shown to target host translation as a virulence strategy (Mohr and Sonenberg 2012 Cell Host Microbe. 12(4):470-83), not much is known in this regard about *Chlamydia* infection. We have now added this discussion to the text on lines 332-336.

The proteomics pipeline and the various parameter cutoffs used for the analysis makes sense. Considering the limited sensitivity encountered in proteomics, it is good to see the overlap of the interferon related and immune pathways between proteomics and transcriptomics.

Overall, apart from a few details (regarding the number of replicates) and the significance of the non-common differentially expressed genes, I find the approach and the results presented

in the transcriptomics and proteomic analysis satisfactory.

REVIEWER #3 (Remarks to the Author):

The paper by Yeung et al., demonstrates the advantages of using iPSC technologies for the study of Chlamydia pathogenesis. The authors showed that macrophages derived from iPSCs (iPSdMs) could be infected with Chlamydia. In addition, they genetically modified iPSCs using CRISPR/Cas9 to knockout IRF5 and IL10RA and found that lack of these genes increased susceptibility of iPSdMs to Chlamydia infection. RNAseq and proteomic analysis has been performed to determine how iPSdMs respond to Chlamydia.

Overall, paper demonstrated the value and advantages of iPSC technologies for study of Chlamydia pathogenesis. However, it would be nice to see how the response of iPSdM to Chlamydia infection is different from somatic macrophages.

Actually it was important to our paper to demonstrate that the response of iPSdM to *Chlamydia* infection was similar to that of somatic macrophages as described above. However there are two important differences now mentioned in response to the comments of Reviewers 1 and 2. First as mentioned there is the issue of macrophage class since it is well known that M1 macrophages can kill *Chlamydia* [Gracey E. et al. 2013 PLoS One. 8:e69421; Bushacher T. et al. 2015 PLoS One. 10:e0143593]. Thus we have now pointed out at lines 37-44, 105-7 and 437-8 that our macrophages resemble M0 to M2 like macrophages that are permissive for *Chlamydia*. Second we did see some transcriptional differences related to metabolic processes and extracellular matrix organization, likely reflecting the development of iPSdM in culture, as discussed in lines 341-349.

During development, macrophages arise from different waves of hematopoiesis. In mouse system Myb-independent and Myb-dependent waves of embryonic hematopoiesis were identified. Authors should comment whether their differentiation protocol recapitulate Myb-dependent or Myb-independent macrophage pathway.

In fact it has been shown that [Vanhee S et al. 2015 Haematologica 100:157-66] *in vitro* human embryonic stem cell hematopoiesis mimics MYB-independent yolk sac hematopoiesis as now mentioned in the text. We compared iPSdMs with human blood derived macrophages since the latter are derived in an analogous manner from progenitor cells and represent the most common model for studying ex vivo macrophage biology. This is now mentioned in the text on lines 89-91.

Does knockout of IRF5 and IL10R affect the yield of macrophages?

No, both knockout mutant iPSCs differentiated (and now with the additional clones, both clones of each knockout) differentiated into similar numbers of macrophages as the parental iPSC line. We have added this information into the Results section lines 377-378.

Supplementary Figure 3d should include data from wild type to allow comparison wild and knockout iPSdMs.

This data has now been added.

Materials and methods should provide references describing iPSC lines used in studies or indicate the method used for iPSC- generation (lenti, episomal, Sendai?).

This has now been added. The HipSci Initiative used the CytoTune 1 Sendai method to reprogram skin tissues from normal human volunteers into iPSCs. The Certificate of Analysis for the hiPSCs are available on the project website (www.hipsci.org).

REVIEWER #4 (Remarks to the Author):

Yeung et al. describe a system for editing iPSCs using CRISPR, then differentiating into

macrophages, which are used as an *in vitro* model for studying *Chlamydia* infections. As the authors have pointed out, the use of iPSC-derived, genetically modified macrophages may prove to be a useful system for studying host-pathogen interactions in general, and in the case of studying *Chlamydia* infections this approach appears to be novel.

As far as method development goes, the editing of iPSC cells with CRISPR – including the use of selection markers to increase efficiency of editing – is fairly well established. The authors' claim of having "not only improved the frequency of biallelic mutations, but also greatly simplified the final genotyping step of the mutant clones by requiring only Sanger sequencing, and is thus particularly useful for the generation of biallelic mutants at scale" is not all at substantiated by the data presented here, and in fact an irresponsible claim to make. The only data presented here, in Figure 6 and Supplementary Data Figure 1 give no indications of the general efficiencies of editing the iPSCs with or without the selection marker, or number of sub-clones assayed and their genotypes.

We have reduced the emphasis on the methods here since, as the reviewer mentions the method of using CRISPR to perform genome editing, is well established. We have also deleted the paragraph in Discussion discussing CRISPR technology. We have responded to the issues below by adding data into the Supplemental section

There are several big problems here:

- *The PCR assay in Supp Fig 1, its unsuitability for genotyping aside, is essentially uninterpretable: what are the lanes? Where are the controls?*

As mentioned above we have confirmed the genotypes of the two IRF5 mutants through PCR, sequencing and RNA-Seq. Fig. S1a has been amended to include a schematic of where the specific PCR primers PF/GR1 and GF1/ER bind and the expected sizes of the PCR bands.

- *The use of Sanger sequencing for genotyping is also problematic. It's fundamentally a population-level assay, and easily masks any heterogeneity in not truly clonal populations.*

We agree with this in principle although our mutants were indeed clonal. The use of Sanger sequencing to genotype our knockout mutants had been validated in a recently accepted paper in *Development* from our co-author W. Skarnes with S. Pollard [Bressan R.B. et al. 2017 *Development*. in press], where they had generated knockout mutations in more than 10 transcription factors and validated their mutants by Sanger sequencing. As part of their validation, selected mutants that gave frameshift, mosaic, or wt clonal Sanger sequencing traces (as shown below) were subjected to HiSeq. Results from HiSeq matched what was observed from the Sanger sequencing. However since we sequenced both clones of the two mutants that were propagated and used these clones to derive macrophages we believe this addresses the issues of heterogeneity.

- *In the IRF5 clone, is the 96bp deletion in frame? And if so does it truly abolish the activity of IRF5?*

As mentioned above we made a mistake in Fig. S1b IRF5 that should indicate a 97bp

deletion rather than a 96bp deletion obviating the potential for an in-frame deletion. The newly added clone also has a frame shift (2bp insertion).

- Have the authors looked at whether the selection marker has integrated into other sites in the genome, or whether Cas9 editing may have affected the expression of any other gene (i.e. all kinds of possible off-target effects)? Without having done a careful genotyping analysis it's really not appropriate to be drawing any functional conclusions.

Yes we confirmed that the selection marker was only present by PCR and also present at a single unique site using qPCR copy number assay (now mentioned in the text line 240-1, and this was one of the criteria for proceeding with particular subclones.

REVIEWERS' COMMENTS:

Reviewer #1 (Remarks to the Author):

The authors responded adequately to all my points and have included additional controls in their revised version of the manuscript. I think the manuscript has strongly been improved and iPSC as infection models for Chlamydia infection will for sure advance the field. Thomas Rudel

Reviewer #2 (Remarks to the Author):

[No further comments for author.]

Reviewer #3 (Remarks to the Author):

Almost all my comments are adequately addressed. However, authors should explain why their macrophages are M0 or M2 type. They provide references to prior studies, but these studies do not discuss the type of obtained macrophages. Authors should describe characteristics of their cells that justify the conclusion that iPSC-derived macrophages are M1 or M2 type.

Reviewer #4 (Remarks to the Author):

Yeung et al. have improved the clarity of Supplementary Figure 1 and added some text to clarify potential off-target issues (though relatively little data are given still on the efficiency of editing or off-target analyses overall). Since the manuscript has de-emphasized any claims on new CRISPR method development, these points are no longer a focus in the study, and as such the CRISPR-related points presented here are satisfactory.